# An Ensemble Docking Approach for Analyzing and Designing Aptamer Heterodimers Targeting VEGF_165_

**DOI:** 10.3390/ijms25074066

**Published:** 2024-04-05

**Authors:** Yeon Ju Go, Mahroof Kalathingal, Young Min Rhee

**Affiliations:** 1Department of Chemistry, Korea Advanced Institute of Science and Technology (KAIST), Daejeon 34141, Republic of Korea; koyj0212@postech.ac.kr; 2Department of Chemistry, Pohang University of Science and Technology (POSTECH), Pohang 37673, Republic of Korea

**Keywords:** VEGF, aptamer heterodimer, ensemble docking, molecular dynamics simulation

## Abstract

Vascular endothelial growth factor 165 (VEGF_165_) is a prominent isoform of the VEGF-A protein that plays a crucial role in various angiogenesis-related diseases. It is homodimeric, and each of its monomers is composed of two domains connected by a flexible linker. DNA aptamers, which have emerged as potent therapeutic molecules for many proteins with high specificity and affinity, can also work for VEGF_165_. A DNA aptamer heterodimer composed of monomers of V7t1 and del5-1 connected by a flexible linker (V7t1:del5-1) exhibits a greater binding affinity with VEGF_165_ compared to either of the two monomers alone. Although the structure of the complex formed between the aptamer heterodimer and VEGF_165_ is unknown due to the highly flexible linkers, gaining structural information will still be valuable for future developments. Toward this end of accessing structural information, we adopt an ensemble docking approach here. We first obtain an ensemble of structures for both VEGF_165_ and the aptamer heterodimer by considering both small- and large-scale motions. We then proceed through an extraction process based on ensemble docking, molecular dynamics simulations, and binding free energy calculations to predict the structures of the VEGF_165_/V7t1:del5-1 complex. Through the same procedures, we reach a new aptamer heterodimer that bears a locked nucleic acid-modified counterpart of V7t1, namely RNV66:del5-1, which also binds well with VEGF_165_. We apply the same protocol to the monomeric units V7t1, RNV66, and del5-1 to target VEGF_165_. We observe that V7t1:del5-1 and RNV66:del5-1 show higher binding affinities with VEGF_165_ than any of the monomers, consistent with experiments that support the notion that aptamer heterodimers are more effective anti-VEGF_165_ aptamers than monomeric aptamers. Among the five different aptamers studied here, the newly designed RNV66:del5-1 shows the highest binding affinity with VEGF_165_. We expect that our ensemble docking approach can help in de novo designs of homo/heterodimeric anti-angiogenic drugs to target the homodimeric VEGF_165_.

## 1. Introduction

The vascular endothelial growth factor (VEGF) family of proteins plays key roles in regulating physiological vasculogenesis and angiogenesis, namely the process of new blood vessel formations [1,2,3]. At the same time, the VEGF family is also responsible for pathological angiogenesis in diseases such as tumor growth and neovascular age-related macular degeneration [2,3,4,5,6,7]. Therefore, VEGF’s family members are important targets for diagnosing these diseases [2,3]. The human VEGF family comprises five members [7], among which the most studied one is VEGF-A, which is often referred to simply as VEGF. It has several isoforms with different numbers of amino acid residues [7], but VEGF_165_ with 165 residues is the most abundant and is known to play an important role in pathological angiogenesis [7]. It is a homodimeric protein, with each monomer having a heparin-binding domain (HBD) and a receptor-binding domain (RBD) connected by a flexible linker (Figure 1A) [7,8]. The experimental structures of HBD and RBD are known [9,10], but due to the high flexibility of the linker region, the full structure of the homodimeric VEGF_165_ is experimentally unknown [8]. Functionally, VEGF binds with VEGF receptors (VEGFRs) and induces dimerization and activation of VEGFR. This triggers signal transduction pathways that are crucial for angiogenesis (Figure 1B) [11,12]. While there are a series of VEGFR variants, VEGFR-2 is known to play a main role in both physiological and pathological angiogenesis [13]. Up to now, significant research efforts have been directed toward discovering inhibitors targeting VEGF to prevent its binding to VEGFR [14]. Several VEGF inhibitors have currently been approved for anti-angiogenic treatment, such as Bevacizumab [15], Aflibercept [16], and Macugen [17].

Aptamers are short single-stranded DNA or RNA molecules used for the molecular recognition of targets with high affinity and specificity [18,19]. Not surprisingly, aptamers have also been used to inhibit the activity of VEGF. Indeed, Macugen is a Food and Drug Administration (FDA)-approved RNA aptamer used for treating age-related macular degeneration [17]. Several DNA aptamers, including VEa5, 2G19, and Vap7, have been further developed as anti-angiogenic agents against VEGF [20,21,22,23]. In addition, efforts have been made to enhance the binding affinity of aptamers to VEGF by designing G-quadruplex-forming aptamers [23,24,25] and aptamer dimers [20,23,26,27,28,29]. Naturally, it is important to find an aptamer with high binding affinity to VEGF. Experimentally, it has been shown that homodimers made for monomeric aptamers VEa5, del5-1, and 3R02 show higher affinities with VEGF_165_ than their respective monomers [26,29]. Another experimental study demonstrated the efficacy of an aptamer heterodimer composed of DNA aptamers V7t1 and del5-1 (V7t1:del5-1) [23]. In particular, V7t1:del5-1 showed higher affinity towards VEGF_165_ than its monomeric components, with del5-1 attaching to the HBD part and V7t1 attaching to the RBD part [23]. V7t1 is a 25-mer aptamer with a G-quadruplex structure and was derived from Vap7 [23]. On the other hand, del5-1 is a 50-mer aptamer derived from VEa5, and it encompasses three stem-loop regions [26]. In the heterodimer, V7t1 and del5-1 are connected using 10 thymine nucleotides as a linker [23]. The structure of the bound complex formed by the aptamer heterodimer and VEGF_165_ is not known experimentally, primarily due to the high flexibility of the linkers present in both components. Even still, understanding the interactions between the aptamer heterodimer and VEGF_165_ will be indispensable for developing new aptamer heterodimers with improved properties.

In the absence of experimental structures, in silico methods such as molecular docking and molecular dynamics (MD) simulations are useful tools [30,31,32]. The molecular docking approach facilitates the prediction of a ligand-binding site on the target molecule as well as the binding affinity and provides useful information on the position and orientation of the ligand [30]. Conversely, MD simulations capture the dynamic behavior of the ligand–target complex, providing useful information on stability, flexibility, and the associated conformational changes [31]. Additionally, end-point free energy methods [32,33,34] have been extensively utilized for calculating ligand–target binding free energies. They often use only the conformations in the free and bound states of the ligand and target and can be computationally less expensive than more rigorous alchemical- and pathway-based free energy methods [35,36,37]. Even though the end-point methods lack somewhat in accuracy compared to these rigorous methods, they can still offer higher accuracy than docking scoring functions [32].

With the above in mind, here we propose an ensemble docking approach for predicting an ensemble of complex structures of V7t1:del5-1 bound to VEGF_165_ by taking into account both small- and large-scale motions in both VEGF_165_ and the aptamer heterodimer. We first utilize anisotropic network model (ANM) analysis to gain insights into large-scale motions based on the lowest-frequency normal modes [38], followed by biased MD simulations to generate molecular structures considering these motions. In fact, identifying large-scale changes that typically involve low-frequency normal-mode motions in a complex system is a challenging task with atomistic MD. ANM can be a reasonable approach to generating large-scale collective motions occurring between the domains/monomers connected by a linker in both VEGF_165_ and the aptamer heterodimer. Through biased MD simulations, we obtain several conformations of VEGF_165_ and the aptamer heterodimer corresponding to their lowest-frequency normal modes. The obtained structures are then used for extraction based on ensemble docking, followed by unbiased MD simulations as well as binding free energy calculations based on molecular mechanics generalized Born surface area (MM/GBSA) [32], to predict the complex structures of the aptamer heterodimer bound to VEGF_165_.

We also designed a new aptamer heterodimer, RNV66:del5-1, against VEGF_165_ by replacing V7t1 with a locked nucleic acid (LNA) modified version of V7t1, namely RNV66. Although V7t1 effectively targets the RBD part of VEGF_165_, it exhibits polymorphism, resulting in multiple G-quadruplex structures [24]. In contrast, the 25-mer RNV66 generated by replacing guanine residues at positions 5, 21, and 24 of V7t1 with LNA-G residues has a single stable G-quadruplex structure [24]. Moreover, RNV66 by itself is an outstanding anti-VEGF aptamer and is known to inhibit cancer proliferation with higher binding affinity and nuclease resistance than V7t1 [25]. Using our ensemble docking protocol mentioned above, we also provide the complex structures of RNV66:del5-1 bound to VEGF_165_. For comparison, the three constituting monomeric units (V7t1, RNV66, and del5-1) are also included in the ensemble docking approach with VEGF_165_. All five VEGF_165_/aptamer complexes obtained are then analyzed in detail to explore binding poses, hydrogen bond (H-bond) interactions in VEGF_165_/aptamer complexes, and steric clashes between the aptamer and VEGFR-2 when the complex interacts with VEGFR-2. In addition, the stability of the G-quadruplex structure of RNV66 or V7t1 upon binding with VEGF_165_, either as a monomer or as part of a heterodimer, is also examined.

## 2. Results and Discussion

### 2.1. Designing Anti-VEGF_165_ Aptamers

In this paper, we study two aptamer heterodimers, RNV66:del5-1 and V7t1:del5-1, as well as three monomeric aptamers, RNV66, V7t1, and del5-1, for targeting VEGF_165_. Through computational means, we tried to incorporate small- and large-scale motions into the consideration of the protein–aptamer interaction [39] involving VEGF_165_.

To have a pictorial sense of the large-scale motion, the lowest-frequency normal mode of VEGF_165_ was first predicted using ANM analysis on VEGF_165_. As mentioned above, this approach was selected because observing large-scale motions induced by flexible linkers solely through unbiased MD simulations is highly challenging due to the sampling difficulty. This motion could be characterized as a scissoring bending vibration between the two HBD units, with oscillations in the distance between the two (Figure 2A). To reflect this large-scale motion, we generated seven different VEGF_165_ conformations by using biased MD simulations, increasing the distance between the centers of mass (COMs) of the two HBD units from 3 nm to 9 nm in 1 nm intervals (Figure 2B). In addition, from each 10 ns long biased MD trajectory, VEGF_165_ snapshots were extracted at every 500 ps to account for small-scale motions of the protein. Accordingly, 20 VEGF_165_ structures were extracted from each biased MD simulation, resulting in a total of 140 (7 × 20) VEGF_165_ conformations. With these, we calculated all possible intra-monomer COM distances between RBD and HBD as well as the inter-monomer ones, and the distances ranged 2.0–6.2 nm and 2.5–6.1nm, respectively (Figure 2A).

With these distance values in mind, we also performed ANM analyses on the equilibrated structures of RNV66:del5-1 and V7t1:del5-1 as obtained from the MD simulations. From these, wavy motions with which the two aptamer monomers became closer and farther from each other were commonly observed as the lowest-frequency normal mode in both heterodimers (Figure 3A). To better handle this large-scale motion, we performed biased MD simulations with restraints on the inter-monomer COM distances by increasing the restraining distances from 2.0 nm to 7.5 nm in steps of 0.5 nm intervals. For each COM distance, the biased simulations were continued up to 10 ns, leading to 12 distinct conformations for each heterodimer bearing large-scale motions (Figure 3B). Small-scale motions were then added by extracting snapshots at 500 ps intervals from each 10 ns biased MD simulation, leading to a total of 240 (12 × 20) structures obtained for each heterodimer. As RNV66 or V7t1 binds to RBD [23,24,25] while del5-1 binds to HBD [26], it was necessary to consider distance ranges between RNV66 and del5-1 and between V7t1 and del5-1 that could encompass the RBD-to-HBD distances. This was the reason we chose the heterodimer distance range of 2.0–7.5 nm. Because V7t1:del5-1 was effective with a linker made of 10 thymine nucleotides [23], we used the same linker for RNV66:del5-1 as well. For the cases of the 3 aptamer monomers, we similarly extracted snapshots at every 500 ps of a single 10 ns MD simulation of each without any restraint, resulting in 20 monomer structures for each monomer type.

### 2.2. Ensemble Docking of Aptamers with VEGF_165_

To generate the VEGF_165_/aptamer complexes, we utilized the HDOCK docking program [40,41] by utilizing all combinations of the conformations described in the previous section. In fact, HDOCK has been extensively adopted recently to analyze protein–RNA and protein–DNA interactions [42,43,44], and it has shown excellent performance [45] in the community-wide Critical Assessment of Prediction of Interactions (CAPRI) [46]. To further verify its applicability to our systems, we conducted a docking experiment between VEGF_121_ and domains 2 and 3 (D23) of VEGFR-2 with it. The docked structure (Appendix A) was a close match with the experimentally known VEGF-A/VEGFR-2 D23 structure (PDB ID: 3V2A) [47] after adding the missing residues in the crystal structure through homology modeling with SWISS-MODEL [48].

In total, 240 structures for each aptamer heterodimer and 20 for each aptamer monomer underwent docking with 140 VEGF_165_ structures. Consequently, a total of 33600 docked poses or structures (i.e., 140 × 240) for each aptamer heterodimer and 2800 poses (i.e., 140 × 20) for each aptamer monomer were obtained (Figure 4). Based on the obtained docking results, we extracted the initial structures required for subsequent MD simulations to calculate the binding free energies with MM/GBSA [32]. For each type of aptamer heterodimer, 400 docked structures (20 VEGF_165_ snapshots × 20 aptamer snapshots) were generated for each of the 84 variants (7 VEGF_165_ variants × 12 aptamer variants). Subsequently, the docked structures that did not meet the “attachment criteria,” i.e., RNV66 or V7t1 attaching to the RBD part and del5-1 attaching to the HBD part, were manually excluded. From the remaining docked structures for each type of aptamer heterodimer, the top-3 complexes with the highest docking scores were selected for each of the 84 variants. Similarly, for each type of monomeric aptamer, 400 docked structures (20 VEGF_165_ snapshots × 20 aptamer snapshots) were generated for each of the 7 variants (7 VEGF_165_ variants × 1 aptamer). Within the docked structures for each type of monomeric aptamer, those that did not satisfy the attachment criteria were eliminated first, and the top-three complexes with the highest docking scores were selected from the rest for each of the seven variants. Consequently, we obtained 252 complex structures each for VEGF_165_/RNV66:del5-1 and VEGF_165_/V7t1:del5-1 and 21 complex structures each for VEGF_165_/RNV66, VEGF_165_/V7t1, and VEGF_165_/del5-1 (Figure 4). Thus, a total of 567 VEGF_165_/aptamer structures were selected as initial structures for the subsequent MD simulations. The docking scores of these selected conformations are provided in Appendix A.

### 2.3. Binding Affinity of Aptamers with VEGF_165_

To obtain binding free energies using MM/GBSA, we first performed 15 ns MD simulations for each complex with the chosen structures. After analyzing the time course of the root-mean-square deviation (RMSD) for each complex from its 15 ns MD trajectory, we confirmed that all 567 structures were equilibrated within 10 ns. The time course of the RMSD for each complex was calculated with respect to its initial structure. However, upon observing the complex structures after the 15 ns period, we found that some complexes did not meet the attachment criteria mentioned above. Thus, these trajectories were excluded from the list, and there remained 491 trajectories in total for further analyses, as follows: 218 for VEGF_165_/RNV66:del5-1; 210 for VEGF_165_/V7t1:del5-1; 21 for VEGF_165_/RNV66; 21 for VEGF_165_/V7t1; and 21 for VEGF_165_/del5-1. For each MD trajectory, using the snapshots sampled at every 10 ps over its last 5 ns, we calculated the binding free energy using the single-trajectory approach of MM/GBSA [32]. The time courses of the RMSD values obtained from the MD trajectories with the lowest binding free energies for the five types of VEGF_165_/aptamer complexes are shown in Figure 5. In this case, the RMSD values were calculated using the complex structure at the 10 ns mark of each MD trajectory as the reference structure.

The average binding free energies for the five types of complexes are listed in Table 1, along with the number of trajectories adopted for generating the averages. The averages revealed that RNV66:del5-1 is the strongest binder and is much better than V7t1:del5-1. In general, heterodimers are better than monomers. Although any quantitative interpretation with specific numbers should be avoided due to the inherent limitation of MM/GBSA, these binding free energies are also in the same trend with the average docking scores also listed in Table 1 and are consistent with available experimental data that showed that V7t1:del5-1 exhibits a higher binding affinity to VEGF_165_ compared to either V7t1 or del5-1 [23]. Similarly, RNV66:del5-1 demonstrates a higher binding affinity than its monomeric counterparts, further supporting the notion that aptamer heterodimers are more effective anti-VEGF_165_ aptamers than monomers. Interestingly, from our results, RNV66:del5-1 appeared as a more potent binder than V7t1:del5-1, suggesting that an aptamer heterodimer designed with RNV66, which is known to be more effective in inhibiting than V7t1 [25], may potentially serve as a superior anti-VEGF_165_ inhibitor. To verify that we are not misguided by any outlying conformation, we selected the structures with the top-ten lowest binding free energies for each of the five types of VEGF_165_/aptamer complexes and re-calculated the averages based only on those top-ten contributors (Appendix A). The results are still consistent with the ones observed with all structures, as discussed above.

When we visually inspected the final snapshots of the 491 VEGF_165_/aptamer trajectories using VMD [49], we noticed some typical binding poses, and we classified them into three categories: sandwich, side, and hug poses. A sandwich pose refers to a configuration where the two HBD units surround the aptamer in the center. In other words, for the VEGF_165_/aptamer complex to exhibit a sandwich pose, both HBD units of VEGF_165_ must bind to a single aptamer domain (Figure 6A). Additionally, in the side pose, del5-1 binds to only one HBD of VEGF_165_, while RNV66 or V7t1 binds to the outside of VEGF_165_ (Figure 6B). In a hug pose, RNV66 or V7t1 is bound to the interior of VEGF_165_, while del5-1 binds to only one of the two HBD units, and the other unit is located far from del5-1 (Figure 6C). In the classification, to assess whether a specific aptamer has bound to a particular domain of VEGF_165_, we considered whether the distances between protein side-chain heavy atoms and DNA heavy atoms were within 0.45 nm [50]. More extensive pictorial representations than those in Figure 6 can be found in Appendix A.

Upon computing the fraction of occurrences of these three binding poses, we observed that all the five aptamers we adopted preferred the sandwich pose when binding to VEGF_165_ (Figure 7). This indicates that the sandwich pose is the most stable structure for complexation. Interestingly, aptamer heterodimers displayed fewer sandwich and hug poses compared to monomeric aptamers. The binding poses of the top-ten complex structures for each type of VEGF_165_/aptamer complex can be found in Appendix A.

### 2.4. Hydrogen Bonds between Aptamer and Key Residues of VEGF_165_

From experiments, the key VEGF_165_ residues involved in its binding with heparin [9] and VEGFR-2 [47] are known. For comparison, we investigated to identify which of these key VEGF_165_ residues are involved in binding with the five aptamers through H-bonds. We focused specifically on examining the formation of H-bonds with the seven residues (Y21, Y25, I43, N62, D63, E64, and Q89) in the RBD part, which were identified as the key residues for binding with VEGFR-2 [47], as well as the ten residues (R123, R124, K125, K140, R145, R149, R156, K162, R164, and R165) in the HBD part, known to interact with heparin (Figure 8) [9].

We adopted the MD trajectories corresponding to the top-ten complex structures with the lowest binding free energies, mentioned in an earlier section, by taking the snapshots at every 10 ps during the last 5 ns of each trajectory and counting the number of H-bonds in each snapshot. The criterion for a H-bond was a maximum donor–acceptor distance of 3.5 Å and a maximum H-donor–acceptor angle of 30 deg [51,52]. We then calculated the time average of the number of H-bonds over the 5 ns period for each trajectory. The results are shown in Figure 9 separately for the RBD residues and the HBD residues. In general, HBD is more prone to forming H-bonds than RBD, and heterodimers tend to form more H-bonds. In a sense, this is not surprising, as we already observed that RNV66:del5-1 and V7t1:del5-1 are more effective binders than their constituting monomers. Out of the 17 key VEGF_165_ residues, R123, R145, R149, R156, K162, R164, and N62 were the most involved in forming H-bonds, frequently with the five aptamers.

### 2.5. Hydrogen Bonds between Aptamer and the Other Residues of VEGF_165_

We also conducted an analysis of H-bond formations involving the other residues of VEGF_165_ to understand why heterodimer aptamers exhibit higher binding affinity compared to monomeric aptamers. This analysis specifically focused on the top-ten structures based on the binding free energy for each VEGF_165_/aptamer complex. For RNV66 and V7t1, the interactions predominantly involved VEGF_165_ residues D35, Q37, R56, and H99 when forming H-bonds with the RBD part. For del5-1, significant involvements were observed with R123, R124, K125, K136, S138, K140, Y142, D143, S144, R145, R149, N154, R156, R159, R164, and R165 in the HBD part. In contrast, the two heterodimer aptamers (RNV66:del5-1 and V7t1:del5-1) formed H-bonds not only with D35, Q37, R56, and H99 but also with K48, K84, H86, and Q89 in the RBD part (Figure 10). In the HBD part, in addition to the important VEGF_165_ residues mentioned for del5-1, interactions with K147, Q150, and K163 were also observed. These aspects illustrate that the higher binding affinities of the heterodimer aptamers result from their ability to bind to both the RBD and the HBD units of VEGF_165_, providing a distinct advantage in binding. We also speculate that homodimer aptamers will likely bind only to either RBD or HBD, and they will have lower binding affinities with VEGF_165_ than the heterodimers.

### 2.6. Steric Clashes between Aptamer and VEGFR-2

We believe that an aptamer can act as an effective inhibitor of VEGFR-2 activation because it can block the VEGF_165_-to-VEGFR-2 interaction itself. To fulfill this purpose, there should be a considerable steric clash between VEGFR-2 and the VEGF_165_/aptamer complex. To estimate this aspect, we employed the PISA server [53] to calculate the total area of the sterically clashing regions between VEGFR-2 and the aptamer based on the final snapshots from the MD trajectories of the top-ten complex structures with the lowest binding free energies. A detailed description is provided later in the Materials and Methods section. Here again, aptamer heterodimers generally exhibited significantly larger areas of steric clashes with VEGFR-2, further supporting the superior effectiveness of the heterodimers (Table 2).

### 2.7. Stability of G-Quadruplex Structures

RNV66 and V7t1 are known to possess G-quadruplex structures [24]. We attempted to see whether the G-quadruplex structures of RNV66 and V7t1 were maintained during their binding to VEGF_165_, either as a monomeric aptamer or as part of a heterodimer. We adopted the trajectories of the VEGF_165_/aptamer complexes containing either RNV66 or V7t1, for a total of 470 trajectories each 15 ns long, as explained in Section 2.3. In the case of RNV66, to assess the stability of a G-quadruplex structure, we calculated the RMSD using the twelve residues that constitute the G-quadruplex region with respect to its NMR structure (PDB ID: 2M53) [24]. For V7t1, we adopted the corresponding twelve residues for the RMSD calculation, and the reference was the energy-minimized structure of V7t1, namely the energy minimization result that started from the NMR structure of RNV66 after replacing LNA-G residues with DNA-G residues. The results are shown in Figure 11, in which it can be observed that the binding of RNV66 and V7t1 to VEGF_165_ either as a monomeric aptamer or as part of a heterodimer did not significantly alter their G-quadruplex structures. Interestingly, RNV66 preserved the G-quadruplex structure better than V7t1, and the extent of the preservation was larger in the monomer case (Figure 11B). This aspect is consistent with the fact that the LNA modifications in RNV66 enhance the stability of the G-quadruplex structure [24,54] and further supports our prediction that RNV66:del5-1 will likely work better than V7t1:del5-1.

## 3. Materials and Methods

### 3.1. Modeling DNA Aptamers

The initial structure of RNV66 was taken from its NMR structure deposited in the Protein Data Bank (PDB ID: 2M53) [24]. Because the experimental structure of V7t1 (5′-TGTGGGGGTGGACGGGCCGGGTAGA-3′) was not available, we generated it from the structure of RNV66 (5′-TGTGLGGGTGGACGGGCCGGLTALA-3′) by replacing its LNA-G residues at positions 5, 21, and 24 with DNA-G residues. This was achieved by removing O2 and C6 atoms, followed by adding single hydrogen atoms to C2 and C4 (Figure 12). Two K^+^ ions were placed between the three G-quartet planes to help maintain the G-quadruplex structures (Appendix A).

In the case of del5-1, due to the absence of an available three-dimensional (3D) structure, it was generated from its sequence information (5′- ATACCAGTCTATTCAATTGGGCCCGTCCGTATGGTGGGTGTGCTGGCCAG-3′) using various nucleic acid folding programs. The energetically most stable secondary DNA structure of del5-1 that can be written in a two-dimensional (2D) form was created with its sequence information using the RNAstructure web server [55]. In fact, the same 2D structure was obtained when we adopted the Mfold web server [56], further validating the structure. However, because predicting a 3D DNA structure based on 2D DNA structure information is not practically feasible, we instead generated a 3D RNA structure first based on the 2D DNA structure information after the apparent nucleobase replacements [57] using the RNAComposer web server [58]. The 3D DNA structure of del5-1 was then obtained from the 3D RNA structure by inversely replacing uracil with thymine. This indirect computational method, including the generation of a 3D DNA structure from a 3D RNA structure, while it is approximate and should be used with care, has been utilized in various earlier studies [59,60]. To generate the V7t1:del5-1 structure, V7t1 and del5-1 were linked by using ten thymine nucleotides. The RNV66:del5-1 heterodimer was generated in a similar manner. Structures, including the RNV66 obtained from the Protein Data Bank and those modeled through various steps, such as V7t1, del5-1, RNV66:del5-1, and V7t1:del5-1, underwent further refinement through energy minimization using GROMACS 2022 [61]. The minimization process was halted when the maximum force convergence reached below 1000 kJ mol^−1^ nm^−1^. Subsequently, 2 ns MD simulations under NPT conditions were conducted for equilibration purposes.

### 3.2. Modeling VEGF_165_

For the protein side, the full experimental structure of the homodimeric VEGF_165_ is not available. However, the X-ray crystal structure of the homodimeric RBD (PDB ID: 2VPF) [10] and the NMR structure of the monomeric HBD (PDB ID: 1VGH) [9] are available, and we manually linked HBD to each RBD monomer using the sequence information [8,39]. The interdomain linker sequence was RPKKDRARQENP, where RPKKD and ARQENP corresponded to the X-ray crystal structure of RBD (PDB ID: 2VPF) and the NMR structure of HBD (PDB ID: 1VGH), respectively. The missing residue R110 was generated using Avogadro [62]. Employing VMD [49], the individual components were manually linked to form the complete structure of the homodimeric VEGF_165_. Following this, the structure was refined through 500 steps of energy minimization and 2 ns of MD simulation under NPT conditions. We stress that the structure after 2 ns MD should not be considered a representative equilibrium structure. Because VEGF_165_ with the linker will inevitably be very flexible at room temperature, it will exist by forming a diverse structural ensemble. This is actually the most important reason for using the ensemble docking method.

### 3.3. Anisotropic Network Model (ANM) Analysis

The lowest-frequency normal modes of the aptamer heterodimers and VEGF_165_ were analyzed using the ANM web server [63]. ANM is a simple normal-mode analysis method at the residue level, showing the collective motions of large molecules [38]. While we utilized ANM to identify the lowest-frequency normal modes of VEGF_165_ and aptamer heterodimers (Figure 2 and Figure 3), we conducted additional 500 ns MD simulations for them to confirm that the motions from ANM appear consistently. The rationale behind this additional endeavor is the fact that both VEGF_165_ and aptamers are quite flexible without any uniquely defined representative equilibrium structures, and how ANM will behave without a well-defined equilibrium structure may be rather unclear. Upon applying ANM to the resulting structures of these long MD simulations, we confirmed that, like the initial structures we employed, the lowest-frequency normal modes were scissoring motions for VEGF_165_ and wavy motions for the aptamer heterodimers.

### 3.4. Docking with HDOCK

All docking calculations were carried out using HDOCK [40,41], with which a target was immobilized and only the ligand was allowed to wander in translational and rotational space with a fixed step size. An interval of 15 deg was used for the rotational sampling, and an interval of 0.12 nm was used for the fast Fourier transform (FFT)-based translational sampling [40]. A shape-based pairwise scoring function was employed to score the binding modes obtained through the sampling. The top-ten translations with the best shape complementarity were re-scored and optimized for each rotation by an iterative knowledge-based scoring function. The best-scored translation was then kept for each rotation [40].

### 3.5. MD Simulations

In the case of unbiased MD simulations, systems were solvated with TIP3P water [64] with a minimum distance of 10 Å between the solute and the edge of the box. Namely, for the initial structures, VEGF_165_ was placed in a simulation box with dimensions of 9.9 × 11.2 × 11.0 nm^3^; RNV66:del5-1 and V7t1:del5-1 were placed in a box with dimensions of 10.9 × 8.7 × 9.5 nm^3^; RNV66 and V7t1 were placed in a box with dimensions of 5.5 × 6.3 × 5.9 nm^3^; and del5-1 was placed in a box with dimensions of 6.9 × 7.9 × 7.8 nm^3^. For charge neutralization, counter ions were added. The DNA aptamers were described using the AMBER OL15 force field [65], with the missing parameters for the LNA-G residues in RNV66 taken from an earlier work [54]. The force field parameters for VEGF_165_ were taken from AMBER ff19SB [66]. The ions were modeled using the parameters proposed by Joung and Cheatham [67]. The topology files for the simulations were generated using AmberTools22 [68]. For each system, energy minimization using the steepest descent method [69] was first taken, followed by equilibration for 2 ns under NPT conditions. The minimization convergence was declared when the maximum force became smaller than 1000 kJ mol^−1^ nm^−1^.

For biased MD simulations, the conformations after 2 ns of initial equilibration were solvated in TIP3P water with a minimum distance of 60 Å between the solute and the edge of the box. With this, the simulation box sizes were the following: for VEGF_165_, 20.6 × 20.1 × 20.3 nm^3^; for RNV66:del5-1, 18.2 × 17.7 × 16.9 nm^3^; and for V7t1:del5-1, 19.2 × 17.0 × 19.7 nm^3^. To generate discrete conformations of VEGF_165_ and aptamer heterodimers along the lowest-frequency normal modes, biasing potentials with a force constant of 1000 kJ mol^−1^ nm^−2^ were applied along the distance between COMs. At this stage, the umbrella sampling tool [70] was employed as a practical tool for conducting biased MD simulations. The duration of any simulation was 10 ns after undergoing energy minimization and equilibration. For each monomeric aptamer that did not require a biased simulation, to be fair, 10 ns of MD simulations were performed without adding any bias. These processes were performed for all the complexes extracted based on the docking scores for further MD simulations, and production MD simulations were performed for 15 ns under NPT conditions, following energy minimization and 100 ps of equilibration under NVT conditions.

All simulations were performed with periodic boundary conditions using GROMACS 2022 [61]. The SHAKE algorithm was used to constrain the bonds involving hydrogen atoms, allowing a time step of 2 fs [71]. The short-range Lennard-Jones interactions were truncated at a cutoff distance of 1.2 nm, and long-range dispersion corrections were applied for energy and pressure. The long-range electrostatic interactions were treated with the Particle Mesh Ewald (PME) approach [72] with a real-space cutoff of 1.2 nm. The temperature was maintained at 300 K using a velocity-rescale thermostat [73] with a relaxation time of 1 ps. The pressure was isotropically coupled at 1 bar employing a c-rescale barostat [74] with a coupling constant of 1 ps and a compressibility of 4.5 × 10^−5^ bar^−1^.

### 3.6. MM/GBSA Binding Free Energy

The MM/GBSA method [32] is an end-point free energy calculation method that maintains a good balance between computational efficiency and accuracy [75]. In particular, it has been widely used for the re-scoring of docked poses in structure-based drug design by calculating binding free energies from MD simulations of the docked poses [76]. In the MM/GBSA method, the binding free energy of a ligand–target complex is calculated as the sum of Δ*E*_MM_, Δ*G*_sol_, and −*T*Δ*S*, where they represent the changes in the gas-phase molecular mechanics (MM) energy, the solvation free energy, and the conformational entropy upon ligand–target binding, respectively [32]. Δ*E*_MM_ is calculated based on the force field used for the simulations, while the polar and nonpolar components of Δ*G*_sol_ are calculated, respectively, using the generalized Born (GB) model and the solvent-accessible surface area (SASA)-based approach [77,78]. The −*T*Δ*S* term is usually calculated either by a normal-mode analysis or by a quasi-harmonic analysis [79], which are computationally burdensome and tend to bear a large margin of errors [80,81,82]. Therefore, MM/GBSA binding free energies are often calculated by ignoring the −*T*Δ*S* term [25,32,80,81,82], especially when the relative binding free energies of closely related ligands with the same target are needed, as in our case [32]. We adopted the single-trajectory MM/GBSA approach [32], which ignores ligand and target conformational changes upon binding, leading to a significant reduction in noise in the binding free energy calculation [82]. All MM/GBSA calculations were performed using AmberTools22 [68].

### 3.7. Total Area of Steric Clashes between Aptamer and VEGFR-2

To estimate the total area of steric clashes between aptamer and VEGFR-2 when the VEGF_165_/aptamer complex binds to VEGFR-2, we first overlaid the final snapshot of the VEGF_165_/aptamer complex obtained after 15 ns of MD simulations onto the X-ray crystal structure of the VEGF-A/VEGFR-2 D23 complex (PDB ID: 3V2A). The structural overlay was performed by aligning the RBD part of the VEGF_165_/aptamer complex with that of the VEGF-A/VEGFR-2 D23 complex, which resulted in significant steric clashes between the aptamer and VEGFR-2. The buried surface area between the aptamer and VEGFR-2 was then calculated using the PISA server [53]. This value was equivalent to the total area of steric clashes between the aptamer and VEGFR-2.

## 4. Conclusions

The aptamer heterodimer V7t1:del5-1 has been shown experimentally to be a superior inhibitor of VEGF_165_ compared to its monomeric counterparts, V7t1 and del5-1 [23], but the flexible linker present in both VEGF_165_ and the aptamer heterodimer is preventing the experimental structure determination of the complex formed between the aptamer heterodimer and VEGF_165_. To overcome this limitation, an ensemble docking approach, considering both small- and large-scale motions in both VEGF_165_ and the aptamer heterodimer, is proposed here to obtain trustworthy structures of the aptamer heterodimer complexed with VEGF_165_. This ensemble docking approach was used to find the complex structures of V7t1:del5-1 and the newly designed RNV66:del5-1, as well as their monomeric counterparts (V7t1, RNV66, and del5-1) with VEGF_165_. The binding free energy analysis found that RNV66:del5-1 has the highest binding affinity with VEGF_165_ among the five types of aptamers studied, suggesting RNV66:del5-1 as a promising new aptamer heterodimer against VEGF_165_. Additionally, aptamer heterodimers show higher binding affinities to VEGF_165_ compared to their monomeric counterparts, consistent with the experimental results. This adds reliability to our ensemble docking approach for generating the structures of VEGF_165_/aptamer complexes. It was found that all five types of aptamers predominantly prefer a sandwich pose while forming complexes with VEGF_165_. In the sandwich pose, the aptamer is sandwiched between the two HBD units of VEGF_165_. Side and hug poses were also observed in the structures of VEGF_165_/aptamer complexes. Compared to monomeric aptamers, aptamer heterodimers have more H-bonds with the key residues of VEGF_165_ involved in the heparin and VEGFR-2 binding events, as well as a larger area of steric clashes with VEGFR-2 when the VEGF_165_/aptamer complex interacts with VEGFR-2, suggesting aptamer heterodimers as the most effective anti-VEGF_165_ aptamers over monomeric aptamers. Based on our simulation results, we found that the G-quadruplex structure of RNV66 or V7t1 is not significantly damaged upon binding with VEGF_165_, either as a monomeric aptamer or as part of an aptamer heterodimer. Moreover, the G-quadruplex structure of RNV66 was found to be more stable than that of V7t1 in the complexes of both monomeric and heterodimeric aptamers with VEGF_165_. The structural stability of RNV66 over V7t1 found in our simulations is consistent with the findings from previous experimental and molecular simulation studies that demonstrated that the presence of LNA residues in RNV66 stabilizes its G-quadruplex structure [24,54]. We believe that our ensemble docking approach, which incorporates both small- and large-scale motions, will be valuable in the development of new homo/heterodimeric therapeutic drugs against the homodimeric VEGF_165_.

## Figures and Tables

**Figure 1 ijms-25-04066-f001:**
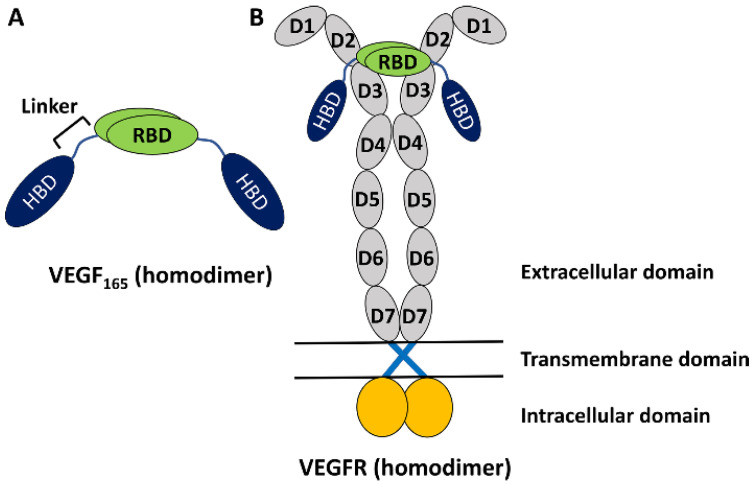
(**A**) Schematic representation of VEGF_165_ as a homodimer with two monomers. Each monomer is composed of RBD and HBD, connected by a flexible linker. (**B**) VEGF_165_ binding with VEGFR toward its activation. VEGFR is also a homodimer and is composed of an extracellular domain with seven sub-domains in each monomer, a transmembrane domain lying in lipid, and an intracellular domain extruding into the cytoplasm.

**Figure 2 ijms-25-04066-f002:**
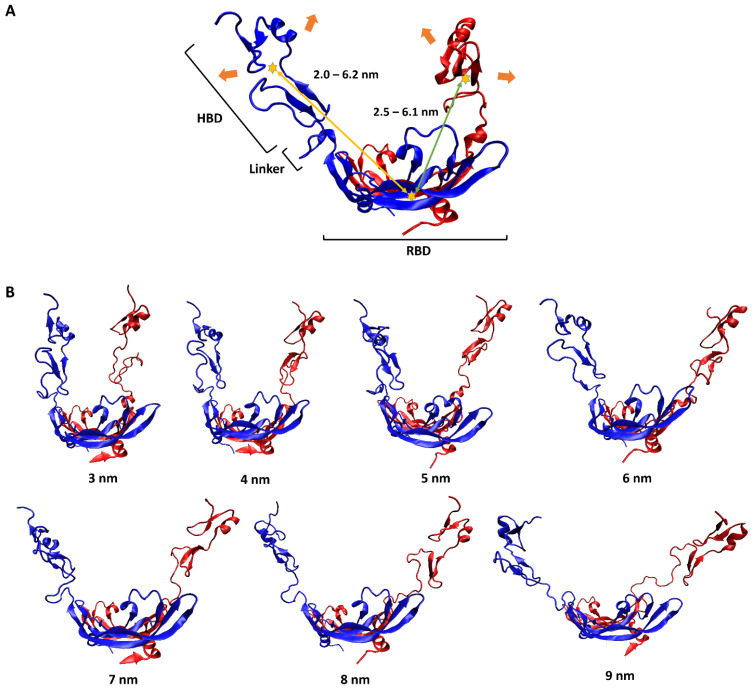
(**A**) Schematic representation of VEGF_165_ motion. The two monomers of VEGF_165_ are shown in blue and red. The orange arrows indicate the lowest-frequency normal-mode motion obtained from ANM. The yellow stars represent the COMs of HBD and RBD, with the yellow and green arrows indicating, respectively, the intra-monomer and inter-monomer COM distances between HBD and RBD. (**B**) Biased MD simulation results for generating seven VEGF_165_ conformations, with HBD–HBD COM distances restrained at the designated distances.

**Figure 3 ijms-25-04066-f003:**
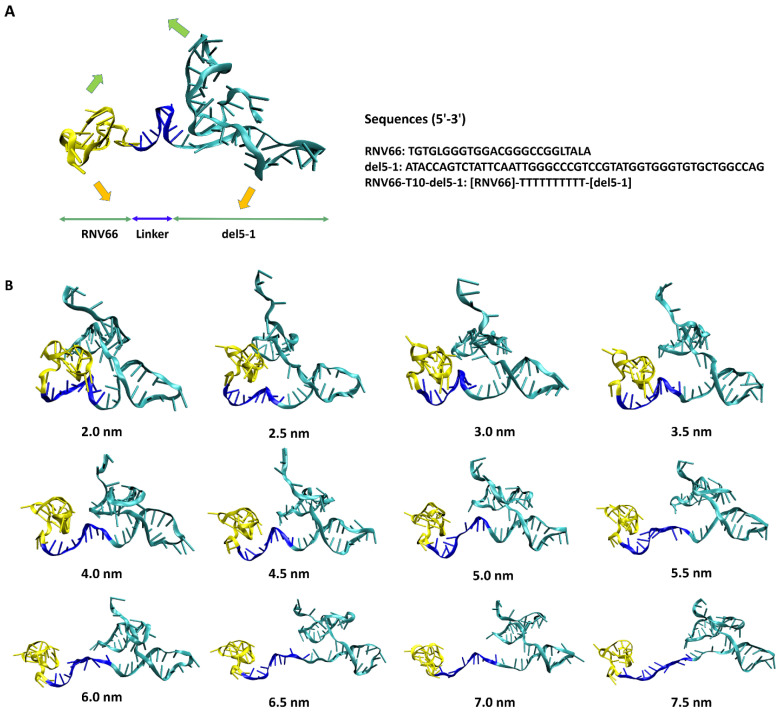
(**A**) RNV66:del5-1 together with its lowest-frequency normal-mode motion from ANM, illustrated with green and orange arrows. The sequence information is also given on the right. (**B**) Twelve RNV66:del5-1 structures obtained from biased MD simulations, with the inter-monomer COM distances restrained at the designated distances.

**Figure 4 ijms-25-04066-f004:**
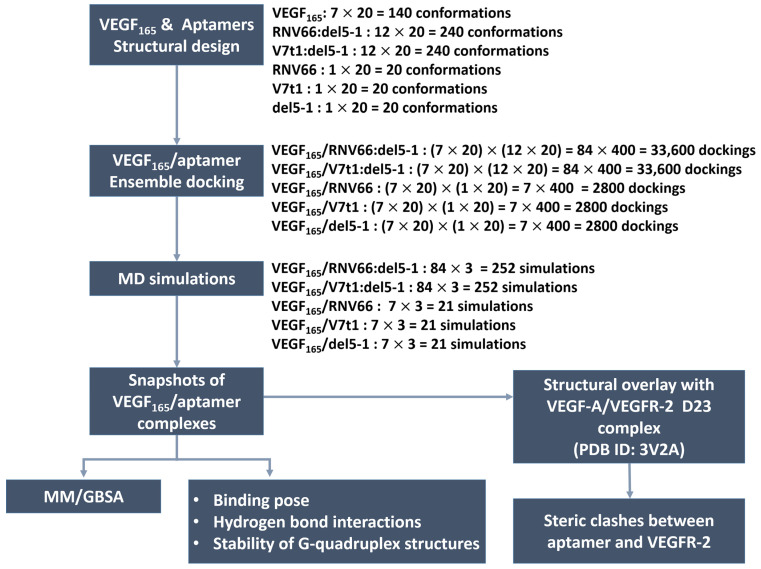
Procedures of our ensemble docking.

**Figure 5 ijms-25-04066-f005:**
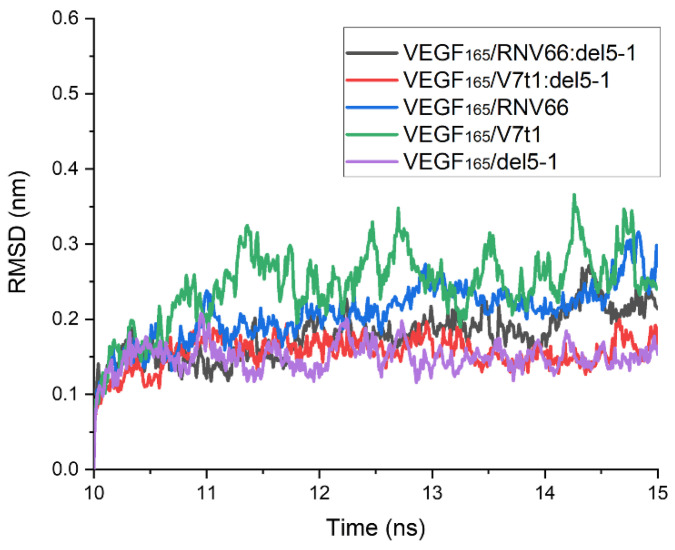
Time courses of RMSD values for the five types of VEGF_165_/aptamer complexes. RMSD values are presented for the 10–15 ns part of the 15 ns long MD trajectory with the lowest binding free energy for each complex. The reference structures for measuring RMSDs were the ones at the 10 ns time point.

**Figure 6 ijms-25-04066-f006:**
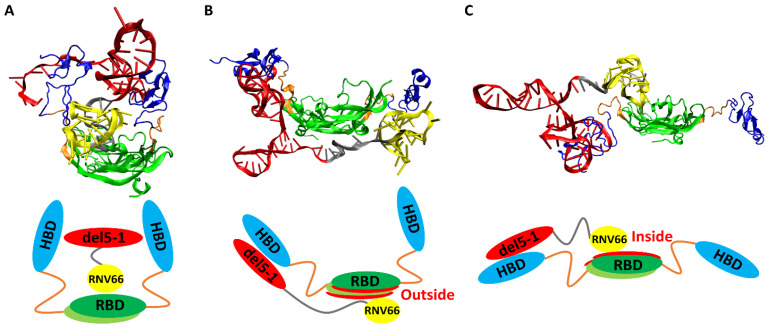
Typical poses of VEGF_165_/aptamer complexes: (**A**) a sandwich pose of VEGF_165_/RNV66:del5-1, (**B**) a side pose of VEGF_165_/RNV66:del5-1, and (**C**) a hug pose of VEGF_165_/RNV66:del5-1. The structures obtained through VMD for each pose are presented alongside their schematic representations. The representative structures for each pose were arbitrarily chosen. For aptamers, RNV66 and V7t1 are shown in yellow, while del5-1 is in red, with the linker in silver. For VEGF_165_, RBD is in green, while HBD is in blue, with the linker in orange.

**Figure 7 ijms-25-04066-f007:**
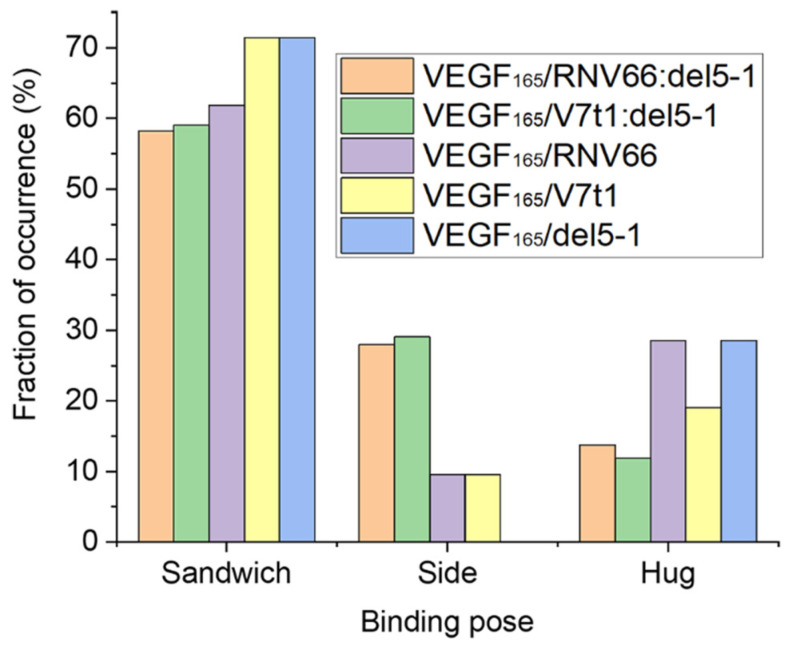
Fraction of occurrences of the sandwich, the side, and the hug poses for each type of VEGF_165_/aptamer complexes.

**Figure 8 ijms-25-04066-f008:**
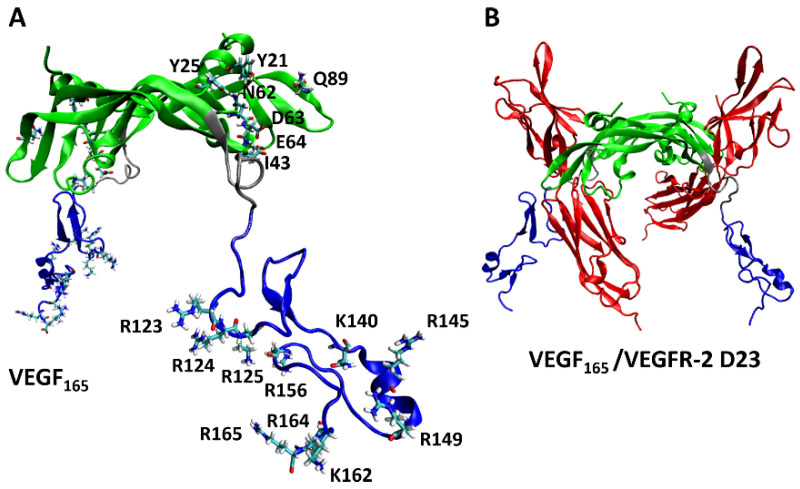
(**A**) Key residues of VEGF_165_ in binding with VEGFR-2 and heparin, with the RBD/HBD/linker unit depicted in green/blue/silver. (**B**) Schematic representation of the complex formed by VEGF_165_ together with VEGFR-2 D23. VEGFR-2 D23 is highlighted in red.

**Figure 9 ijms-25-04066-f009:**
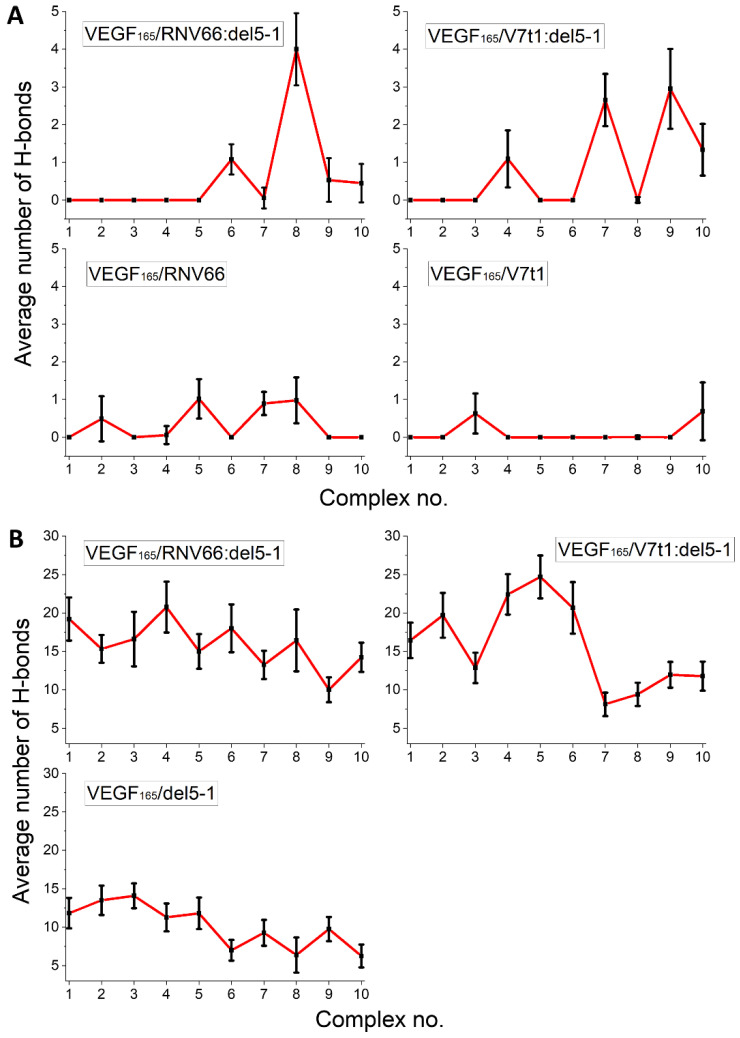
Average number of H-bonds formed between the aptamers and the key residues of VEGF_165_ in (**A**) the RBD and (**B**) the HBD parts. Horizontal axes denote the top-ten complex structures for the given complex type.

**Figure 10 ijms-25-04066-f010:**
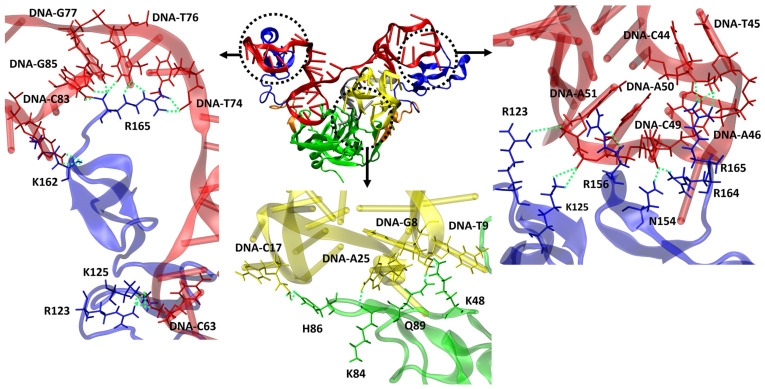
Representation of VEGF_165_/RNV66:del5-1 complex and H-bonds formed between aptamer and VEGF_165_. RNV66 is shown in yellow, del5-1 in red, the linker between RNV66 and del5-1 in silver, the RBD units in green, the HBD units in blue, and the linker between RBD and HBD in orange. The upper panels represent the important residues for forming H-bonds between the HBD part and del5-1, while the lower panel depicts the crucial residues for forming H-bonds between the RBD part and RNV66. H-bonds are depicted as green dashed lines.

**Figure 11 ijms-25-04066-f011:**
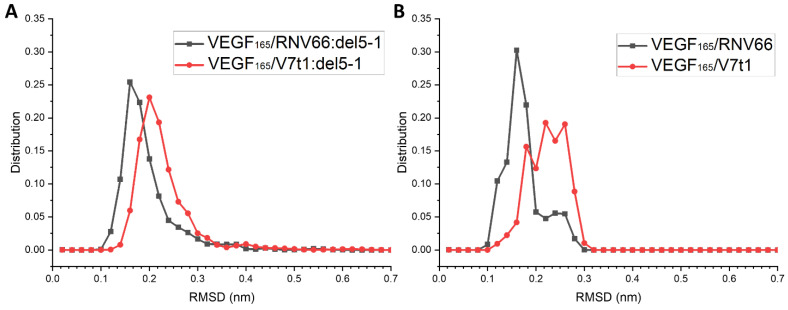
RMSD distributions of the G-quadruplex structures in RNV66 and V7t1 within (**A**) the complexes with heterodimeric aptamers and (**B**) the complexes with monomeric aptamers.

**Figure 12 ijms-25-04066-f012:**
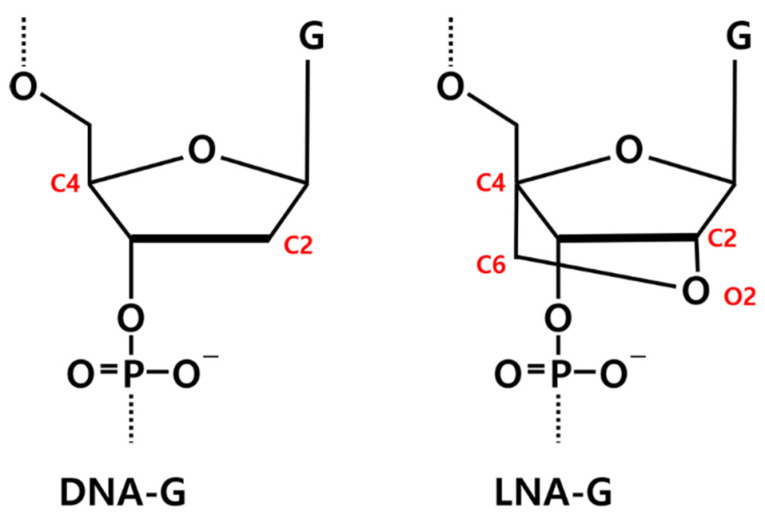
Chemical structures of DNA-guanine (DNA-G) and LNA-guanine (LNA-G). Atoms involved in the LNA bridging are highlighted in red.

**Table 1 ijms-25-04066-t001:** Averaged binding free energies (ΔG) and docking scores for the five types of VEGF_165_/aptamer complexes.

Complex	VEGF_165_/ RNV66:del5-1	VEGF_165_/ V7t1:del5-1	VEGF_165_/ RNV66	VEGF_165_/ V7t1	VEGF_165_/ del5-1
No. of trajectories	218	210	21	21	21
ΔG (kcal/mol)	−199.3 ± 44.5	−117.2 ± 41.0	−47.9 ± 24.9	−74.5 ± 44.1	−93.4 ± 30.1
Docking score ^a^	−1377.8 ± 83.3	−1361.7 ± 76.6	−1094.1 ± 57.5	−1072.5 ± 41.3	−1295.7 ± 71.8

^a^ In an arbitrary unit.

**Table 2 ijms-25-04066-t002:** Total area of steric clashes in nm^2^ between aptamer and VEGFR-2 for the top-10 complex structures of each VEGF_165_/aptamer complex type.

Complex No.	VEGF_165_/ RNV66:del5-1	VEGF_165_/ V7t1:del5-1	VEGF_165_/ RNV66	VEGF_165_/ V7t1	VEGF_165_/ del5-1
1	35.9	12.9	16.3	2.5	62.0
2	25.0	24.5	22.0	16.8	27.1
3	43.9	51.8	9.8	19.2	26.8
4	26.0	50.9	8.5	8.1	23.1
5	39.9	30.6	13.8	16.9	15.2
6	56.8	52.7	6.1	19.2	22.3
7	36.8	43.0	16.9	7.4	27.2
8	68.1	51.4	16.3	15.3	50.0
9	48.9	38.3	14.6	15.3	18.0
10	34.1	47.5	10.6	7.4	11.9
Average	41.5	40.4	13.5	12.8	28.4

## Data Availability

The initially designed structures of the DNA aptamers and VEGF_165_, along with the structures utilized in MD simulations, docking, MM/GBSA calculations, and associated files, i.e., input, parameters, topology files, and scripts for data analyses, are accessible to the public on GitHub (https://github.com/GoYeonju/Ensemble-docking (accessed on 24 November 2023)). The MD simulations, analyses, and MM/GBSA calculations were performed using GROMACS 2022 and AmberTools22, and all docking procedures were performed using HDOCK.

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
