# Peer review of "An Ensemble Docking Approach for Analyzing and Designing Aptamer Heterodimers Targeting VEGF165"

_ijms, 2024, doi:10.3390/ijms25074066_

Round 1

Reviewer 1 Report

Comments and Suggestions for Authors

The manuscript “An ensemble Docking Approach for Analyzing and Designing Aptamer Heterodimers Targeting VEGF165” is devoted to in silico study of Vascular endothelial growth factor 165 (VEGF156), a vasculogenic and angiogenic protein with therapeutic potential in ischemic disorders and tumors. VEGF165, along with VEGF121, is the most abundant isoform of the VEGF-A protein. As the active receptor blocking molecule, the authors chose two heterodimers connected by a flexible linkers: 1) a DNA aptamer heterodimer composed of monomers of V7t1 and del5-1 and 2) an LNA (locked nucleic acid)-modified DNA aptamer heterodimer RNV66 and del5-1. The docking of monomeric units (V7t1, RNV66, and del5-1) is also examined. The work is supported by numerous references to experimental research, was carried out at a high level and will be of undoubted reader interest.

The methods section is complete. The use of methods is justified for the current task. The only remark concerns the short length of the trajectories of 10 ns. The assigned tasks have been solved. The conclusions are justified. The presented problems are of undoubted interest to researchers. The manuscript is recommended for publication after minor revision.

Lines 439-440 using the RNAComposer web server [59].

Which secondary structure prediction method of RNAComposer was used?

Line 448 further refinement through energy minimization.

What package was used to perform the minimization?

Line 467. structure generated by AlphaFold quite resembles the VEGF165 structure we designed 467

The structures predicted by AlphaFold have been reported in various papers to be different from those resolved experimentally. Therefore, they should not be taken as reference. Particularly flexible regions may vary.

4.6. MM/GBSA Binding Free Energy

It looks like some of the symbols have been replaced by bird icons. Line 544 etc. Check also Table 1.

Author Response

Responses to Reviewer’s Comments

First of all, we thank the reviewer for his/her valuable criticism and suggestions. We are convinced that the revisions based on his/her comments have made our discussion clearer. Below, we will list the changes that we have made together with the reviewer’s actual comments in a point-by-point manner. For the reviewer’s convenience, a version of the manuscript with the changes highlighted in yellow is additionally uploaded as a review-only material.

Reponses to Reviewer #1

Lines 439-440 using the RNAComposer web server [59].

Which secondary structure prediction method of RNAComposer was used?

Author reply: For secondary structure prediction, we utilized the RNAstructure web server with the nucleic acid type set to DNA, allowing us to obtain the DNA secondary structure. This DNA secondary structure was then passed to RNAcomposer as a user-provided structure. We could have selected RNAstructure as the secondary structure prediction method within RNAcomposer without providing the DNA-mode generated structure, but then an RNA secondary structure would have been generated without taking DNA into any consideration.

Line 448 further refinement through energy minimization.

What package was used to perform the minimization?

Author reply: In our manuscript, all energy minimization procedures were conducted using the GROMACS 2022 package. We have explicitly indicated this in the text to ensure clear communication of information.

Changed on p. 15: … underwent further refinement through energy minimization.

→ … underwent further refinement through energy minimization using GROMACS 2022.

Line 467. structure generated by AlphaFold quite resembles the VEGF165 structure we designed 467

The structures predicted by AlphaFold have been reported in various papers to be different from those resolved experimentally. Therefore, they should not be taken as reference. Particularly flexible regions may vary.

Author reply: We appreciate for the constructive feedback from the reviewer. Considering the limitations of AlphaFold, especially in accurately representing flexible regions, comparing the structure generated by AlphaFold with our designed structure seems meaningless. Thus, we have removed the section related to AlphaFold from the manuscript.

4.6. MM/GBSA Binding Free Energy

It looks like some of the symbols have been replaced by bird icons. Line 544 etc. Check also Table 1.

Author reply: We sincerely appreciate the reviewer’s careful check. We have corrected all the mistakes now.

Reviewer 2 Report

Comments and Suggestions for Authors

The manuscript represents a computational investigation into the complex formation between vascular endothelial growth factor and two DNA aptamers, employing a suite of computational methodologies, including molecular dynamics simulation and free energy calculations. While the overall methodology is sound, I have the following comments to enhance the manuscript.

1.        The manuscript currently contains excessive verbiage and jargon, frustrating comprehension. The readability would be greatly improved by adopting more precise and formal tone, eliminating unnecessary details, and emphasizing the main findings in each section.

2.        The use of biased simulations for VEGF165 and the DNA aptamers raises questions. The authors first observed large-scale motions in unbiased simulations, why is it necessary to reproduce these motions using biased simulations? Meanwhile, the origin and justification of the chosen distance ranges remain unclear.

3.        The author confirmed the equilibration of their complexes by evaluating the RMSD values. While this is valid for the systems, VEGF165/RNV66:del5-1 and VEGV165/V7t1:del5-1, which has a steady RMSD values below 4Å, it is not applicable to the other three systems, which stabilize around RMSD values of 10Å. This suggests a significant deviation from initial conformations, and it is not indicative if they are or are not still undergoing large-scale conformational changes. Reporting RMSF values could provide a clearer indication of equilibration.

4.        The identification of three binding poses through visual inspection is subjective. A quantitative approach to distinguish these interactions is essential. Furthermore, it is very hard to compare these poses in Figure 6.

5.        For molecular renderings, it is recommended to set the display to orthographic rather than perspective to accurately represent the molecules’ dimensions in 2D.

6.        Some mathematical symbols are not printed correctly, such as in Table 1 and on page 17.

Comments on the Quality of English Language

The manuscript could benefit from more concise expression to enhance clarity and readability.

Author Response

Responses to Reviewer’s Comments

First of all, we thank reviewer for his/her valuable criticism and suggestions. We are convinced that the revisions based on his/her comments have made our discussion clearer. Below, we will list the changes that we have made together with the reviewers’ actual comments in a point-by-point manner. For the reviewer’s convenience, a version of the manuscript with the changes highlighted in yellow is additionally uploaded as a review-only material.

Reponses to Reviewer #2

  1. The manuscript currently contains excessive verbiage and jargon, frustrating comprehension. The readability would be greatly improved by adopting more precise and formal tone, eliminating unnecessary details, and emphasizing the main findings in each section.

Author reply: We are grateful for the valuable advice from the reviewer. Following the reviewer’s suggestion, we have removed unnecessary expressions from the manuscript and strived to convey the essential content of each section more concisely. Introduction was especially heavy with many jargons, and we have eliminated them as much as possible. There are some drug names that might appear as jargons, but using them was unavoidable.

Changed on p. 2: There are three types of VEGFRs in human: VEGFR-1, -2, and -3 [14]. In fact, VEGFR-1 and VEGFR-2 in combination with VEGF-A are the main players in physiological and pathological angiogenesis [14]. As the name suggests, RBD of VEGF is used for binding with VEGFRs [9,12]. The VEGF/VEGFR signaling is mediated by two types of co-receptors, namely neuropilins and heparin-like molecules [15]. In contrast, HBD of VEGF is utilized for binding with its co-receptors [9,15].

→ While there are a series of VEGFR variants, VEGFR-2 is known to play a main role in both physiological and pathological angiogenesis [13].

Changed on p. 3: … We first utilize anisotropic network model (ANM) analysis [41] and biased MD simulations for the purpose. In fact, identifying large-scale changes that typically involve low-frequency normal mode motions in a complex system is a formidable task with atomistic MD, and ANM can be a reasonable tactic toward generating large-scale collective motions occurring between the domains/monomers connected by a linker in both VEGF165 and the aptamer heterodimer.

→ … We first utilize anisotropic network model (ANM) analysis to gain insights into large scale motions based on the lowest frequency normal modes [38], followed by biased MD simulations to generate molecular structures considering these motions. In fact, identifying large-scale changes that typically involve low-frequency normal mode motions in a complex system is a challenging task with atomistic MD. ANM can be a reasonable approach to generating large-scale collective motions occurring between the domains/monomers connected by a linker in both VEGF165 and the aptamer heterodimer.

Changed on p. 3: … Obtaining structural information for either aptamer heterodimer or VEGF165 is experimentally challenging due to the flexibility induced by their linkers. In such a case, computational means may become valuable tools. Indeed, incorporating small- and large-scale motions into the consideration of protein-aptamer interaction is possible [39], and it should form a viable solution toward the in-silico design of a flexible molecule like an aptamer heterodimer together with VEGF165.

→  ... Through computational means, we tried to incorporate small- and large-scale motions into the consideration of the protein-aptamer interaction [39] involving VEGF165.

Other changes: There are many other small changes in multiple sections that we would rather not list here for the succinctness of this response letter. In any case, if the reviewer feels that more revisions are needed, we will be happy to work further.

  1. The use of biased simulations for VEGF165 and the DNA aptamers raises questions. The authors first observed large-scale motions in unbiased simulations, why is it necessary to reproduce these motions using biased simulations? Meanwhile, the origin and justification of the chosen distance ranges remain unclear.

Author reply: Large-scale motions induced by flexible linkers, such as HBD movement, are very expensive to statistically sample. Therefore, actually observing such large-scale fluctuations through unbiased MD simulation becomes extremely challenging. Hence, we initially confirmed large-scale motions of HBD using ANM analysis, not through MD. The purpose was to determine that the lowest frequency normal mode motion of VEGF165 involves scissoring bending vibration by the two HBD units, resulting in oscillations of the distance between the two units. We observed that this flexible motion took place in the distance range of 3 to 9 nm, and adopted it as a criterion for the biased simulation. Namely, to reflect the effect of this lowest frequency normal mode, we generated VEGF165 structures with distances between the two HBD units ranging from 3 to 9 nm using biased MD simulations. To avoid a misunderstanding that we have caused in an earlier version, we have added the following one more sentence to the manuscript.

Added on P. 3-4: As mentioned in the above, this approach was selected because observing large-scale motions induced by flexible linkers solely through unbiased MD simulations is highly challenging due to the sampling difficulty.

  1. The author confirmed the equilibration of their complexes by evaluating the RMSD values. While this is valid for the systems, VEGF165/RNV66:del5-1 and VEGV165/V7t1:del5-1, which has a steady RMSD values below 4Å, it is not applicable to the other three systems, which stabilize around RMSD values of 10Å. This suggests a significant deviation from initial conformations, and it is not indicative if they are or are not still undergoing large-scale conformational changes. Reporting RMSF values could provide a clearer indication of equilibration.

Author reply: The reviewer is right. As noted, small RMSD means structural stability, but large RMSD does not really mean anything. To address this, we recalculated the RMSD using the structure at the 10 ns mark as the reference and confirmed that all systems exhibited RMSD values below 4Å. We have appropriately revised Figure 5 and included relevant information in the main text. We again thank the reviewer for this constructive and careful criticism.

Added on p. 7: In this case, the RMSD values were calculated using the complex structure at the 10 ns mark of each MD trajectory as the reference structure.

Changed on p. 8: The reference structure for measuring RMSD is the one at time zero.

→ The reference structures for measuring RMSDs were the ones at 10 ns time point.

  1. The identification of three binding poses through visual inspection is subjective. A quantitative approach to distinguish these interactions is essential. Furthermore, it is very hard to compare these poses in Figure 6.

Author reply: We appreciate the reviewer’s insightful feedback. Initially, we categorized the VEGF165/aptamer complexes into three primary binding poses through visual inspection. However, to accurately determine the population of these poses, we indeed established quantitative criteria. For the sandwich pose, both HBDs of VEGF165 were required to bind to a single aptamer domain. Conversely, for the side pose, del5-1 was expected to bind to one HBD of VEGF165, while RNV66 or V7t1 needed to bind to the outside of VEGF165. Lastly, for the hug pose, del5-1 should bind to one HBD of VEGF165, while RNV66 or V7t1 should bind to the inside of VEGF165. Additionally, to assess if a specific aptamer effectively binds to a particular VEGF165 domain, we tested whether there were any heavy atoms of protein side-chains within 0.45 nm of any DNA heavy atoms. We have incorporated similar explanation into the manuscript. Regarding Figure 6, to improve the clarity, we have added a schematic diagram to explain the poses.

Changed on p. 9: A sandwich pose refers to a configuration where the two HBD units surround the aptamer in the center (Figure 6A), while a side pose has the aptamer attached to the outside of VEGF165 (Figure 6B). In a hug pose, the aptamer is bound to the interior of VEGF165. In this last case, only one of the two HBD units surrounds the aptamer and the other unit is located far from the aptamer (Figure 6C).

→ A sandwich pose refers to a configuration where the two HBD units surround the aptamer in the center. In other words, for the VEGF165/aptamer complex to exhibit a sandwich pose, both HBD units of VEGF165 must bind to a single aptamer domain (Figure 6A). Additionally, in the side pose, del5-1 binds to only one HBD of VEGF165, while RNV66 or V7t1 binds to the outside of VEGF165 (Figure 6B). In a hug pose, RNV66 or V7t1 is bound to the interior of VEGF165, while del5-1 binds to only one of the two HBD units, and the other unit is located far from the del5-1 (Figure 6C). In the classification, to assess whether a specific aptamer has bound to a particular domain of VEGF165, we considered whether the distances between protein side-chain heavy atoms and DNA heavy atoms are within 0.45 nm [50].

Changed on p. 9: The representative structures for each pose were visually chosen without any quantitative measure.

→ The structures obtained through VMD for each pose were presented alongside their schematic representations. The representative structures for each pose were arbitrarily chosen.

  1. For molecular renderings, it is recommended to set the display to orthographic rather than perspective to accurately represent the molecules’ dimensions in 2D.

Author reply: Following the reviewer’s suggestion, we have revised Figure 10 into the orthographic mode. If there are any further suggestions, we will be happy to further revise figures.

  1. Some mathematical symbols are not printed correctly, such as in Table 1 and on page 17.

Author reply: We sincerely thank the reviewer for his/her careful checks. The typographical errors are now fixed.

Round 2

Reviewer 2 Report

Comments and Suggestions for Authors

The authors have addressed all my comments carefully. I recommend publication in the present form.